# *Lysinibacillus* sp. GG242 from Cattle Slurries Degrades 17β-Estradiol and Possible 2 Transformation Routes

**DOI:** 10.3390/microorganisms10091745

**Published:** 2022-08-30

**Authors:** Sicheng Wu, Peng Hao, Changlong Gou, Xiqing Zhang, Lixia Wang, Wangdui Basang, Yanbin Zhu, Yunhang Gao

**Affiliations:** 1College of Animal Science and Technology, Jilin Agricultural University, Changchun 130118, China; 2College of Animal Science and Technology, Inner Mongolia University for Nationalities, Tongliao 028000, China; 3Northeast Institute of Geography and Agroecology, Chinese Academy of Sciences, Changchun 130102, China; 4Institute of Animal Husbandry and Veterinary Medicine, Tibet Academy of Agricultural and Animal Husbandry Science, Lhasa 850009, China

**Keywords:** *Lysinibacillus* sp., 17β-estradiol, degradation, E2 biotransform routes

## Abstract

Environmental estrogen pollution has long been a concern due to adverse effects on organisms and ecosystems. Biodegradation is a vital way to remove estrogen, a strain of *Lysinibacillus* sp. was isolated, numbered strain GG242. The degradation rate of 100 mg·L^−1^ 17β-estradiol (E2)) > 95% in one week, and compared with extracellular enzymes, intracellular enzymes have stronger degradation ability. Strain GG242 can maintain a stable E2 degradation ability under different conditions (20–35 °C, pH 5–11, salinity 0–40 g·L^−1^). Under appropriate conditions (30 °C, pH 8, 1 g·L^−1^ NaCl), the degradation rate increased by 32.32% in one week. Based on the analysis of transformation products, inferred E2 was converted via two distinct routes. Together, this research indicates the degradation potential of strain GG242 and provides new insights into the biotransformation of E2.

## 1. Introduction

Estrogens are divided into natural and synthetic. Natural estrogens include estrone (E1), 17β-estradiol (E2), and estriol (E3). The main sources of natural estrogen are human and animal excrement, so estrogen inevitably enters the environment, and estrogen is difficult to degrade in nature, resulting in its gradual accumulation. Estrogen can cause many adverse effects, such as the feminization of male fish, and when exposed to E2 concentrations of 550–630 ng·L^−1^, the fish has higher plasma vitellogenin (VTG), gonad somatic index (GSI) reduction, and testicular lesions [1]. Estrogen pollution is widespread in the Yangtze River basin of China, Sichuan and Hunan provinces have high annual emissions of fecal steroid hormones, up to 4789 and 3868 kg, respectively, and Jiangsu province has high E2 content in surface water (16.65 ng·L^−1^) [2]. In the United States, it has been detected in surface waters [3]. The methods of removing estrogen include adsorption, advanced chemical oxidation, photocatalysis, etc. Microbial degradation is seen as a promising removal method due to its environmental friendliness, efficiency, and low cost.

Natural estrogen E2 is noted for its high content and strong activity, many E2-degrading strains have been isolated, for example, five E2-degrading bacteria were isolated from activated sludge, which achieved in vitro a 99% removal rate of 1 mg·L^−1^ E2 within 144 h [4]. Bacteria from deep-sea sediments achieved in vitro about 100% degradation of 5 mg·L^−1^ of E2 within five days [5]. The study of E2 transformation routes can help deeply understand the degradation fate of E2. Some of the reports on E2 transformation routes are summarized in Figure 1.

Here, to degrade E2, a bacterium of *Lysinibacillus* sp. was isolated from cattle farm sewage (solid-liquid separated and unfermented fresh sewage, obtained from Guangze ecological pasture, Changchun, China). (1) Explore the ability to degrade E2 and the E2-degrading capacity of intracellular and extracellular enzymes. (2) Explore the effects of culture conditions (pH, temperature, and salinity) on E2 degradation. (3) Explore the changes in degradation ability under suitable conditions. (4) The E2 transformation products were identified by liquid chromatography tandem electrospray ionization mass spectrometry (LC-ESI-MS) and infer possible transformation routes.

## 2. Materials and Methods

### 2.1. Chemicals and Culture Mediums

E2 and E1 (purity ≥ 98%; 1 g; CAS: 50-28-2; purity ≥ 98%; 100 mg; CAS: 53-16-7) were purchased from Solarbio (Beijing, China). High-performance liquid chromatography (HPLC)-grade acetonitrile (99%, CAS: 75-05-8) and methanol (99%, CAS: 67-56-1) were purchased from Thermo Fisher Scientific (Shanghai, China). Ultrapure water was produced using an ultrapure water device (KMCR-50, Haochun Instrument Equipment, Chengdu, China). The basal salt medium (BSM) was comprised of 1.0 g K_2_HPO_4_, 0.5 g KH_2_PO_4_, 0.2 g MgSO_4_, and 1.0 g (NH_4_)_2_SO_4_ in 1 L ultrapure water. Luria Bertani medium (LB): 10.0 g tryptone, 10.0 g NaCl, 5.0 g yeast extract in 1 L ultrapure water, solid plates were made by adding 1.5% agar to a liquid medium.

### 2.2. Morphology and Growth Curve of Strain GG242

To observe bacterial colony morphology, the enriched strain was streaked on LB plates, a single colony was obtained by repeated streaking, and the single colony was picked for gram staining and microscopic examination (PH100-3B41L-IPL, Phenix, Jiangxi, China). Furthermore, bacteria were inoculated into the LB at 1% (*v*/*v*), sampling at specific time intervals, and detected with the OD_600_ (a UV/visible spectrophotometer MU701, Shimadzu, Kyoto, Japan). Sterilized LB was used as a control. The growth curve was determined by plotting the average value for each time point. The culture was centrifuged (3000 revolutions per minute (rpm), 10 min, 4 °C), the supernatant was discarded, 2.5% glutaraldehyde solution (electron microscope fixative) was added to the pellets and placed at 4 °C for 12 h. After fixing, the mixture was centrifuged (6000 rpm, 5 min, 4 °C), the supernatant was discarded, the pellets were washed 3 times with phosphate-buffered saline (PBS) solution, then centrifuged. Thereafter, the bacteria were dehydrated by sequential exposure to 10, 30, 50, 70, and 90% ethanol, 5 min at each concentration, and then centrifuged (6000 rpm, 5 min, 4 °C). Finally, the bacteria were exposed to 100% ethanol. Scanning electron microscopy was done by the Gechuang Testing Technology Center (Shanghai, China).

### 2.3. Identification of Strain GG242

Strain purity was assured by repeated streaking in LB plates to obtain a single colony, transferring the colony to 5 mL LB and cultured at 30 °C for 24 h. Chromosomal deoxyribonucleic acid (DNA) was extracted from the cultured strain, and DNA was used as a template for 16S rDNA gene sequence amplification via polymerase chain reaction (PCR). 16S rDNA was connected to the T vector to construct a recombinant plasmid, and the vector universal primer was used as the sequencing primer for DNA sequencing. After removing the vector sequence, the gene sequence was obtained. The obtained gene sequences were aligned to other 16S rDNA sequences in the NCBI database, and the sequences with high scores in the aligned result were selected to construct the phylogenetic tree (MEGA7).

### 2.4. Exploring the E2 Degradation Performance of Strain GG242 and the Ability of Intracellular and Extracellular Enzymes to Degrade E2

To explore the E2-degrading ability of strain GG242, the initial concentration of E2 was set to 10, 50, 100, and 200 mg·L^−1^, respectively. First, a single colony was transferred to 5 mL LB and cultured in a shaker (ZWYC-2102C, ZHICHENG, Shanghai, China) at 30 °C and 135 rpm·min^−1^ for 12 h, then 1 mL culture to 100 mL LB, grown to the logarithmic growth phase and centrifuged (7000 rpm, 10 min, 4 °C), supernatant was removed and the pellets were washed twice with PBS solution, centrifuged (7000 rpm, 10 min, 4 °C) after each washing. Finally, the pellets were suspended in PBS solution, and the OD_600_ was adjusted to 2.0 (4.3·10^8^ CFU·mL^−1^). The obtained bacterial suspension was added to BSM for the E2 degradation experiment, the degradation period was 7 days, with daily sampling to detect the OD_600_ and residual E2. To explore the ability of intracellular and extracellular enzymes to degrade E2, bacteria were grown in 100 mL LB to log phase, intracellular and extracellular enzymes were obtained based on the method described in. In brief, the bacterial culture was centrifuged (10,000 rpm, 5 min, 4 °C), the supernatant was used as an extracellular enzyme, the remaining bacteria were resuspended in PBS, and the bacteria were disrupted in ice water bath to obtain intracellular enzymes. The ultrasound cell crusher (VCX130PB, Sonics, Shanghai, China) was used to crush bacteria, the instrument parameter settings were: Power was 150 W, total time was 5 min, working time and interval time were 5 and 7 s, the whole process was repeated 3 times. After the preparation, the enzyme solution was filtered (0.22 μm poly tetra fluoroethylene filters, Jinteng, Tianjing, China), and finally placed at 4 °C. The 5% (*v*/*v*) intracellular enzymes and extracellular enzymes were added to the BSM to degrade E2, respectively, the degradation period was 5 days, and the degradation rate of E2 was used to evaluate the degradation ability.

### 2.5. Exploring E2 Degradation Performance at Different pH, Temperature, and Salinity

As described above, the suspension was prepared, and the OD_600_ was adjusted to 2.0. To explore the effect of different culture conditions on E2 degradation, the following experiments were set up: Group 1: The pH was set to 5–11, group 2: The temperature was set to 20–40 °C, Group 3: The NaCl was set to 0–40 g·L^−^^1^, only one condition was changed in each group, the rest were the same. The degradation period was 5 days, with daily sampling to detect E2. Thereafter, according to the above experiments, the appropriate conditions were selected, and E2 was degraded under appropriate conditions.

### 2.6. Detection of E2 Transformation Products by LC-ESI-MS

To detect the transformation products of E2, the concentration of E2 in the BSM was set to 100 mg·L^−1^ (to ensure sufficient products), the degradation period was set to 5 days, and samples were taken on days 1 and 4 for LC-MS detection. Before detection, 400 μL methanol (−20 °C) was added to 100 μL sample, fully mixed and centrifuged (12,000 rpm, 10 min, 4 °C). The supernatant was concentrated and dried under vacuum. After drying, 150 μL 2-chlorophenylalanine (4 ppm) 80% methanol solution was added to redissolve it, and the supernatant was filtered to obtain the sample to be detected. Chromatographic conditions: ACQUITY UPLC^®^ HSS T3 column (150 × 2.1 mm, 1.8 µm), column oven temperature 40 °C, autosampler temperature 8 °C, injection volume 2 μL, mobile phase for gradient washing was positive ion 0.1% formic acid water (C), 0.1% formic acid acetonitrile (D), negative ion 5 mM ammonium formate water (A), acetonitrile (B), and the flow rate was 25 mL·min^−1^. The setting of the gradient elution program was 0–1 min, 2% B/D; 1–9 min, 2–50% B/D; 9–12 min, 50–98% B/D; 12–13.5 min, 98%B/D; 13.5–14 min, 98–2% B/D; 14–20 min, 2% D positive mode (14~17 min, 2% B negative mode). MS conditions: Electrospray ion source (ESI), positive and negative ion ionization mode. The positive ion spray voltage and the negative ion spray voltage were 3.5 kV and 2.5 kV, respectively, sheath gas 30 arb, and auxiliary gas 10 arb. The capillary temperature was 325 °C, the full scan was performed with a resolution of 70,000, the *m*/*z* scanning range was 81–1000, data-dependent acquisition (DDA) MS/MS experiments were performed with HCD scan, and the collision voltage was 30 eV. Unnecessary MS/MS information was removed with dynamic exclusion (database: BioDeepDB; NoNA; MzCloud; GNPs; MetaDNA).

### 2.7. Statistical Analyses and E2 Detection Methods

Unless otherwise specified, the above degradation experiments were performed in 100 mL BSM at 135 rpm·min^−1^ and 30 °C, bacteria suspension was inoculated with 1% (*v*/*v*), in triplicate, and without bacteria as control. Using Excel (version 2016) to process data and calculate results, statistical analysis of data with SPSS software (applying one-way ANOVA, F-test, and Least Significant Difference test; version 25.0). Graphs were prepared using GraphPad Prism (version 7.00). Detection of E2 by HPLC, HPLC setting conditions: the instrument was Shimadzu LC-2030 Plus (Kyoto, Japan). The chromatographic column was Agilent ZORBAX SB-C18 (250 × 4.6 mm, 5 μm, USA). The mobile phase was acetonitrile: Water = 55:45 (*v*/*v*) with a flow rate of 1.0 mL·min^−1^, column oven temperature 30 °C, injection volume 25 μL, detection wavelength 280 nm.

## 3. Results

### 3.1. Characterization and Identification of the Strain GG242

In cattle farming, steroids and steroid compounds are used to promote growth [11] and high concentrations of E2 were detected in cattle farm sewage [12,13]. Therefore, sewage from cattle farms was selected as the source of bacterial isolation. An E2 degrading bacteria was isolated from the cattle farm sewage, No. strain GG242. The colony was circular, yellow, and smooth surface (Figure 2a). Strain GG242 was a long rod-shaped gram-positive bacterium (Figure 2b,c). The growth curve is shown in Figure 2d, the logarithmic growth phase of the strain was between 4 and 26 h, and the growth of bacteria was gradually stable after 26 h. The 16S rDNA sequence (1516 bp) of the strain GG242 was submitted to GenBank (MZ027481.1). GG242 was closest to *Lysinibacillus* sp.7B-842 (KF441703.1), and the sequence similarity >99% (Figure 3). Therefore, GG242 was classified as *Lysinibacillus* sp.

### 3.2. E2 Degradation by Strain GG242, Intracellular and Extracellular Enzymes

The degrade ability of strain GG242 was assessed by E2 residual concentration and degradation rate in various concentrations of E2 (Figure 4a,b). E2 concentration gradually decreased as the degradation progressed for 10 mg·L^−1^, the residual E2 was below the HPLC detection line on the 3rd day, and as the E2 concentration increased, the remaining E2 gradually increased. For 50 mg·L^−1^ and 100 mg·L^−1^ E2, it was degraded to 1.7 mg·L^−1^ and 3.2 mg·L^−1^ with 96.5% and 96.7% degradation rates on the 7th day. When E2 increased to 200 mg·L^−1^, the residual E2 was 129.2 mg·L^−1^ and the degradation rate was 35.3% (*p* < 0.01) on the 7th day. Furthermore, as E2 was degraded, E1 was detected and accumulated over time (Figure 4c), implying that E1 is a transformation product of E2 and weak ability to degrade the latter. Compared with no E2, adding E2 shows higher biomass (Figure 4d), this may because strain GG242 can use E2 as a carbon source for growth. Similarly, *Novosphingobium tardaugens* ARI-1 can use E2 as a carbon source to support growth [14]. Biomass increased with increasing E2 concentration at 10–100 mg·L^−1^ E2, as the concentration continued to increase, the biomass decreased. The results of E2 degradation by intracellular and extracellular enzymes are shown in Figure 4e. Intracellular enzymes can achieve higher degradation rate during the whole degradation period, and the degradation rate of intracellular enzymes can reach 94.58% on the 1st day and increase slowly as degradation progresses. The degradation rate of extracellular enzymes gradually increased as the degradation progressed, reaching 92.61% on the 3rd day and then slowly increased. In general, both intracellular and extracellular enzymes can degrade E2, and intracellular enzymes show greater degradation ability.

### 3.3. Effects of Temperature, pH, and Salinity on the Degradation Ability of Strain GG242

The E2 degradation results at different pH are shown in Figure 5a,b. At pH 5 and pH 11, the remaining E2 was 14.3 mg·L^−1^ and 12.6 mg·L^−1^ on the 5th day, and the degradation rate (52.2% and 57.8%) was significantly lower than other groups (*p* < 0.01). At pH 6–10, strain GG242 can maintain a stable degradation capacity, the remaining E2 was <3.6 mg·L^−1^ and the degradation rate was >88%. At pH 8, strain GG242 has the strongest degradation ability with 0.3 mg·L^−1^ remaining E2 and 98.9% degradation rate. At different temperatures (Figure 5c,d), as the temperature increased, the remaining E2 gradually decreased and the degradation rate gradually increased. At 20 °C, the remaining E2 was 6.6 mg·L^−1^ with 77.9% degradation rate on the 5th day, as the temperature rose, at 25 °C, the remaining E2 was 1.5 mg·L^−1^ with 94.8% degradation rate. The degradation rate reached the maximum at 30 °C (1.3 mg·L^−1^ E2 and degradation rate >95%). This shows that 30 °C is the preferred temperature for strain GG242. When the temperature increased from 30 to 40 °C, the degradation ability gradually decreased at 35 °C and 40 °C, the remaining E2 was 6.7 mg·L^−1^ and 24.9 mg·L^−1^, with 77.5% and 16.9% (*p* < 0.01) degradation rates. Under different salinity, the degradation rate elevated as the salinity increased (Figure 5e,f). When the NaCl reached 1 g·L^−1^, the degradation capacity was the highest, the remaining E2 was 1.3 mg·L^−1^ with a 95.5% degradation rate on the 5th day. As the NaCl concentration continued to increase, the degradation rate began to decrease and reached a nadir at 40 g·L^−1^ (18.1 mg·L^−1^ remaining E2 and 39.6% degradation rate; *p* < 0.01). Under appropriate degradation conditions, strain GG242 can degrade 200 mg·L^−1^ E2 to about 54 mg·L^−1^ in seven days, and the degradation rate reaches 72.7% (Figure 6). The degradation rate represents a 32.32% improvement (*p* < 0.01) compared to before (40.3%), therefore, in practical applications, optimizing controllable factors will undoubtedly enhance degradation capacity.

### 3.4. The Proposed E2 Transformation Route

Eight transformation products were identified by LC-ESI-MS, identification information, such as retention time and mass–to–charge ratio (*m*/*z*), etc., are shown in Table 1. Chromatogram and mass spectrum are shown in Figure 7. The transformation products can be divided into two categories, one is the cleaved product, and the other is uncleaved. Strain GG242 degraded E2 may proceed through the following two routes, these routes were inferred based on the structures, peak areas, and other reports of the products (Figure 8). On route I, *E2* converted to **2-methoxyestradiol,** next, the rings except for the benzene ring on the 2-methoxyestradiol were cleaved and converted to **trans-Ferulic acid**. On route II, E2 converted to *E1*, and E1 underwent conversion and cleavage. On route II-a, the saturated ring cleavage, formed **4-hydroxycinnamic acid**, 4-hydroxycinnamic acid continued to cleavage and became **hydroquinone**, finally, hydroquinone entered the tricarboxylic acid cycle. On route II-b, the benzene ring on E1 underwent an addition reaction, converted to **dehydroepiandrosterone**, and further converted to **ursodeoxycholic acid***,* ursodeoxycholic acid was subsequently dehydroxylated to form **lithocholic acid**. On route II-c, the benzene ring of E1 was hydroxylated to form **4-hydroxyestrone**, as on routes I and II-a, the saturated ring of 4-hydroxyestrone was cleaved to **3,4-dihydroxymandelic acid**, and further cleaved into **catechol**, then catechol entered the tricarboxylic acid cycle. This study deduces several possible E2 transformation routes, which enriched the fate of E2 in the degradation process and provided new ideas for the biotransformation of E2. Subsequent work will focus on validating these routes and mining potential genes and enzymes involved in the reaction.

## 4. Discussion

In vitro, *Lysinibacillus* have also treated other contaminants, *Lysinibacillus* can produce an extracellular polymer. Within 30 min, 2.5 mg polymer in vitro can absorb 82% of malachite green (100 mg·L^−1^) [26]. In addition, *Pseudomonas* sp. and *Lysinibacillus* sp. can form biofilms on the surface of plastic debris, which contributes to its degradation [27]. However, fewer reports on the degradation of E2, based on this, further research on the strain GG242 was carried out. The strain GG242 can maintain a good degradation capacity at 10–100 mg·L^−1^ E2, the degradation rate is more than 96%, and the first five days were the main phase of degradation, and the degradation capacity was weakened on days 6 and 7, this may be because the bacteria entered a lag phase. At 200 mg·L^−1^ E2, GG242 degraded 46.5 mg·L^−1^ E2, and the degradation rate reached 23.2% on the 1st day, but as the degradation progressed, the degradation rate increased less, on the 7th day, the degraded E2 was 70.8 mg·L^−1^, and the degradation rate only increased to 35.3%. This may be due to two reasons, the first reason may be the high E2 concentration inhibits the degradation ability, and the other may be that the gradually produced E2 metabolite affects the bacteria. With the initial E2 increases, the biomass increases, resulting in a higher degradation rate, this may be due to higher biomass leading to the higher efficiency of biochemical reactions [28]. However, the growth of bacteria began to decline at 200 mg·L^−1^. Although GG242 can use E2 as energy for growth, as E2 concentration gradually increases, E2 may adversely affect bacterial growth. From the analysis of biomass changes, E2 metabolites may influence bacteria, such as at 100 mg·L^−1^ E2, the biomass on the 1st day was comparable to 50 mg·L^−1^ E2, but as degradation proceeded, it showed a sharp drop, which could be that the high concentration of E2 produced more metabolites and affected the growth of bacteria. Compared to other strains, strain GG242 can degrade higher concentrations of E2, such as a *Novosphingobium* strain can degrade about 70% of 100 mg·L^−1^ E2 within 7 days [29]. Zhou et al. [30] isolated a *Nubsella* strain, which can achieve 99% removal of 2 mg·L^−1^ E2 in 7 days. The difference may be due to the different E2 concentrations selected when screening bacteria, strains screened at low E2 concentrations may have higher-affinity enzyme systems, whereas these strains may not have strong estrogen tolerance. Therefore, more studies should be carried out to better evaluate the performance of the two types of bacteria. Both intracellular and extracellular enzymes can degrade E2, and the degradation ability of intracellular enzymes is stronger than extracellular enzymes, this may be due to the mechanism of repairing the activity of endogenous enzymes and proteins [31]. When the degradation rate of the two groups increases to a certain level, the increase of the degradation rate gradually slows down, on the one hand, it may be due to reduced substrate resulting in slower degradation, and on the other hand, it may be due to the low affinity of enzymes for low concentrations of E2. E2 degrading bacteria were isolated, and they degrade different concentrations of E2 under experimental conditions, but degradation capability drops drastically when faced with a harsh environment (in vitro high temperature or low temperature, strong acid or alkali, and high salinity, etc.). Therefore, it is important to evaluate the ability of bacteria to degrade E2 under different conditions. pH, temperature, and salinity are frequently mentioned influencing factors. Strong alkali or acid affect the activity of microorganisms [32,33], and the degradation rate of strain GG242 also decreases at high or low pH. Therefore, appropriate pH is one of the keys for bacteria to function, in addition to affecting microorganisms, pH affects the removal of TCP (2,4,6-trichlorophenol) by ferrate [34]. The effect of temperature on microorganisms is similar, for E2 degrading *Gordonia* sp. strain R9, E2 degradation decreased from 30 to 40 °C [35], and the activity of laccase (estrogen-degrading enzyme) decreases as the temperature rises [36]. The reason for affecting bacteria may be that temperature affects the permeability of membranes [22]. The effect of salt concentration on degradation is also similar, such as a high concentration of NaCl inhibits the degradation of crude oil by soil microbial communities [37], low concentration of NaCl slightly inhibits the degradation of direct blue 15 (DB15) in wastewater, and high concentration strongly inhibit the degradation [38]. This is consistent with the present study, high salinity led to a hypertonic environment, which may be the main factor affecting bacteria. In summary, strain GG242 can maintain stable degradation ability under pH 5–11, 20–35 °C, and 0.1–40 g·L^−1^ NaCl, this indicates that GG242 can adapt to changing environments. Probing the effect of different factors on degradation ability not only shows the potential of bacteria in applications, but also reveals the suitable environment for bacteria.

Two possible routes were deduced based on the analysis of the products, one is the cleavage of E2, and the other is the cleavage and transformation of E1. On routes I and II-a, trans-Ferulic acid and 4-hydroxycinnamic acid have a similar chemical structure, this further illustrates the result of cleavage. Trans-Ferulic acid and 2-methoxyestradiol have the same benzene ring structure, so inferred the two substances are on the same route, but the key compound 2-methoxyestradiol was not identified, which requires further identification. On route II-a, 4-Hydroxycinnamic acid has the same benzene ring structure as E1 and E2. Since E1 is the major transformation product of E2, it is inferred that 4-Hydroxycinnamic acid is the cleavage product of E1. On route II-b, dehydroepiandrosterone can be used to synthesize ursodeoxycholic acid [15], ursodeoxycholic acid and lithocholic acid can be interconverted [20,24]. Therefore, dehydroepiandrosterone conversion reaction is possible, the key point is the conversion of E1 to dehydroepiandrosterone. Dehydroepiandrosterone can be converted to estrogen in humans and animals [23,25,39]. The conversion of E1 to dehydroepiandrosterone has not been reported, so this step needs to be validated to refine route. On route II-c, two substances with a benzene ring structure similar to 4-hydroxyestrone were identified, and the peak area shows a decreasing trend, so inferred 3,4-dihydroxymandelic acid and catechol were the products of 4-hydroxyestrone cleavage, 4-hydroxyesrone had been reported as the product of E1 hydroxylation [6,40]. However, 4-hydroxyesrone was not identified, probably due to the meta-cleavage, and 4-hydroxyesrone can be detected in the presence of 3-chlorocatechol (an inhibitor of meta-cleavage) [6], 2-methoxyestradiol on route Ⅰ was not identified may also be caused by this reason. In the sum of the above analysis, it is inferred that route II is the main transformation route.

## 5. Conclusions

An E2-degrading strain GG242 was isolated from cattle farm sewage and identified as *Lysinibacillus* sp. Strain GG242 has a robust degradation ability, strong and compared with extracellular enzymes, intracellular enzymes have stronger degradation ability. The strain has cultural adaptability, but the elevated temperature has a significant impact on degradation, degradation capacity is improved under suitable conditions (30 °C, pH 8, and NaCl = 0.1 g·L^−1^). Two possible E2 transformation routes were deduced based on the analysis of the product. This study provides an option for estrogen-degrading microorganisms and proposes 2 possible E2 transformation route.

## Figures and Tables

**Figure 1 microorganisms-10-01745-f001:**
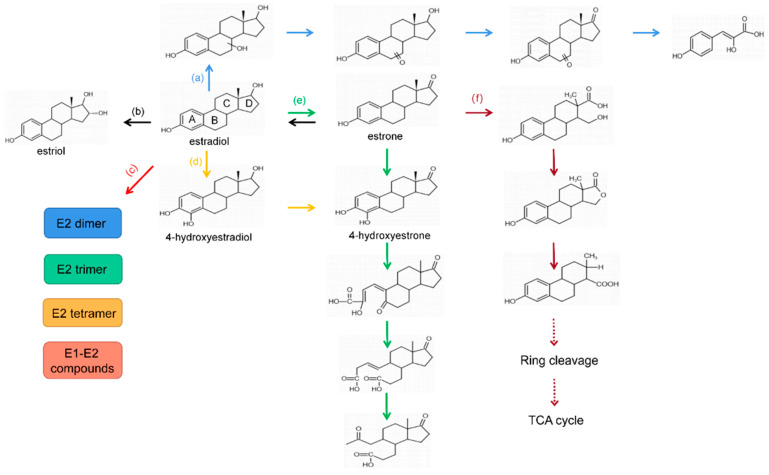
The previously reported 17β-estradiol (E2) biotransformation routes, (a) the saturated ring of E2 was attacked and cleavage [6]; (b) the D ring of E2 was hydroxylated and converted to E3 under the conditions of sulfate-reducing [7]; (c) E2 first became a free radical under the action of laccase, followed by the free radical undergoing a coupling reaction and further transforming into E1, E2 dimer, E2 trimer, E2 tetramer, and E1-E2 cross-coupling products [8]; (d) the benzene ring on E2 was hydroxylated to 4-hydroxyestradiol, and further metabolize [6]; (e) E2 became E1, E1 hydroxylation to 4-hydroxyestrone, and then the benzene ring was cleaved [9]; (f) E2 converted to E1, and the D ring of E1 was cleaved [9], and E1 also converted to E2 under anaerobic conditions [10].

**Figure 2 microorganisms-10-01745-f002:**
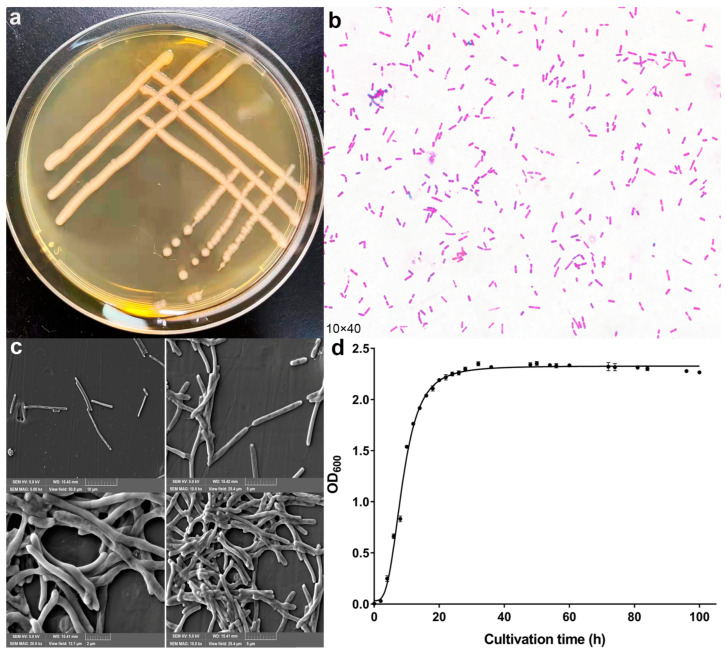
(**a**) Colony morphology on LB plate, (**b**) bacterial morphology after gram stain, (**c**) bacterial morphology under a scanning electron microscope, (**d**) growth curve of strain GG242 in LB medium, data points are the average and error bars represent the standard errors of three experiments.

**Figure 3 microorganisms-10-01745-f003:**
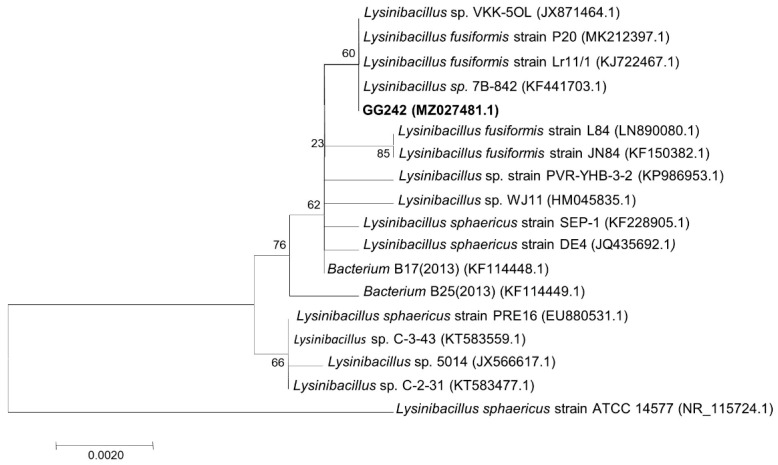
Phylogenetic tree of strain GG242 (the neighbor-joining method based on 1000 bootstraps was used).

**Figure 4 microorganisms-10-01745-f004:**
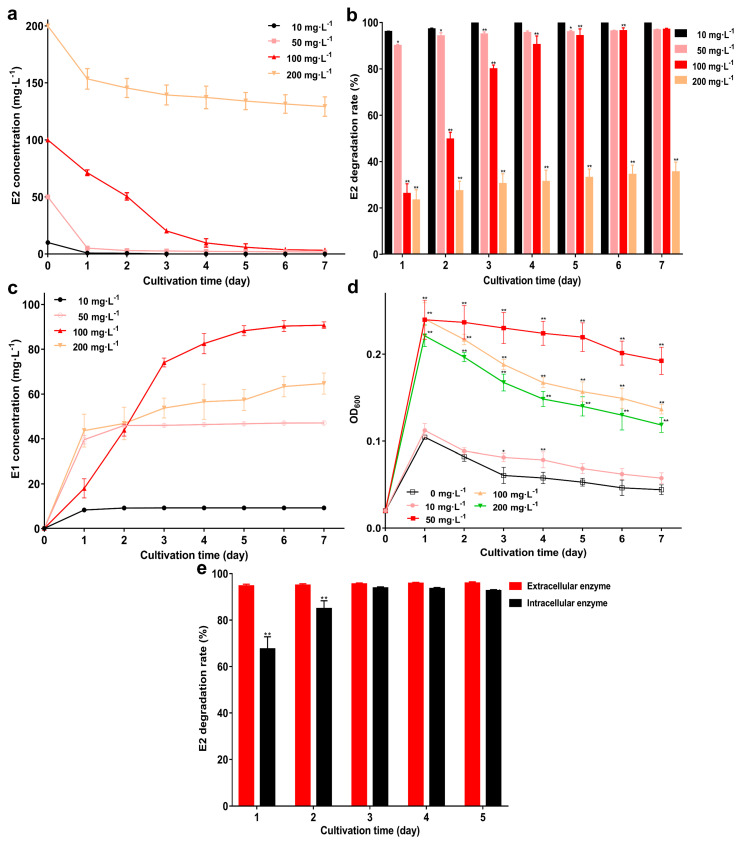
E2 degradation, E1 production, biomass, and IC50. (**a**,**b**) The residual concentration and degradation rate of various concentrations of E2. (**c**) The E1 production at different E2 concentrations; (**d**) Bacterial biomass at different concentrations of E2. (**e**) E2 degradation rate under intracellular enzymes and extracellular enzymes. Data points are the average and error bars represent the standard errors of three experiments * (0.01 < *p* < 0.05); ** (*p* < 0.01).

**Figure 5 microorganisms-10-01745-f005:**
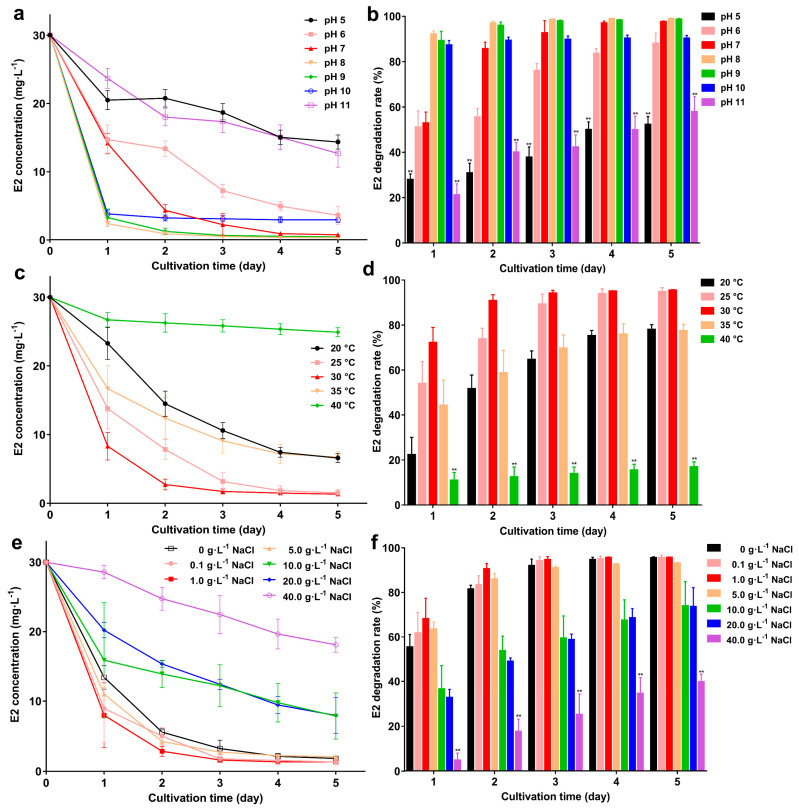
E2 residual concentration and degradation rate under various conditions; (**a**,**b**) pH 5–11; (**c**,**d**) 20–40 °C; (**e**,**f**) 0–40.0 g·L^−1^ NaCl. Data points are the average and error bars represent the standard errors of three experiments; ** (*p* < 0.01).

**Figure 6 microorganisms-10-01745-f006:**
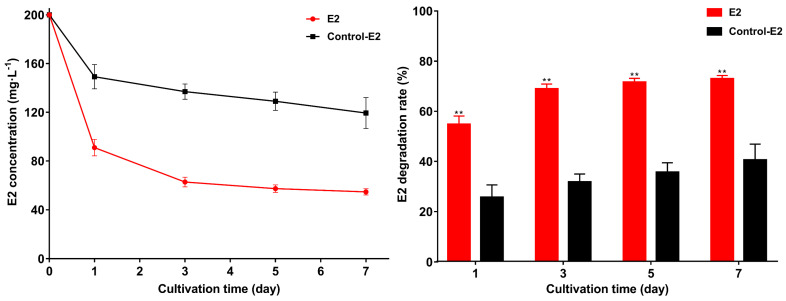
E2 residual concentration and degradation rate under optimized conditions (30 °C, pH 8, NaCl = 1.0 g/L). Data points are the average and error bars represent the standard errors of three experiments; ** (*p* < 0.01).

**Figure 7 microorganisms-10-01745-f007:**
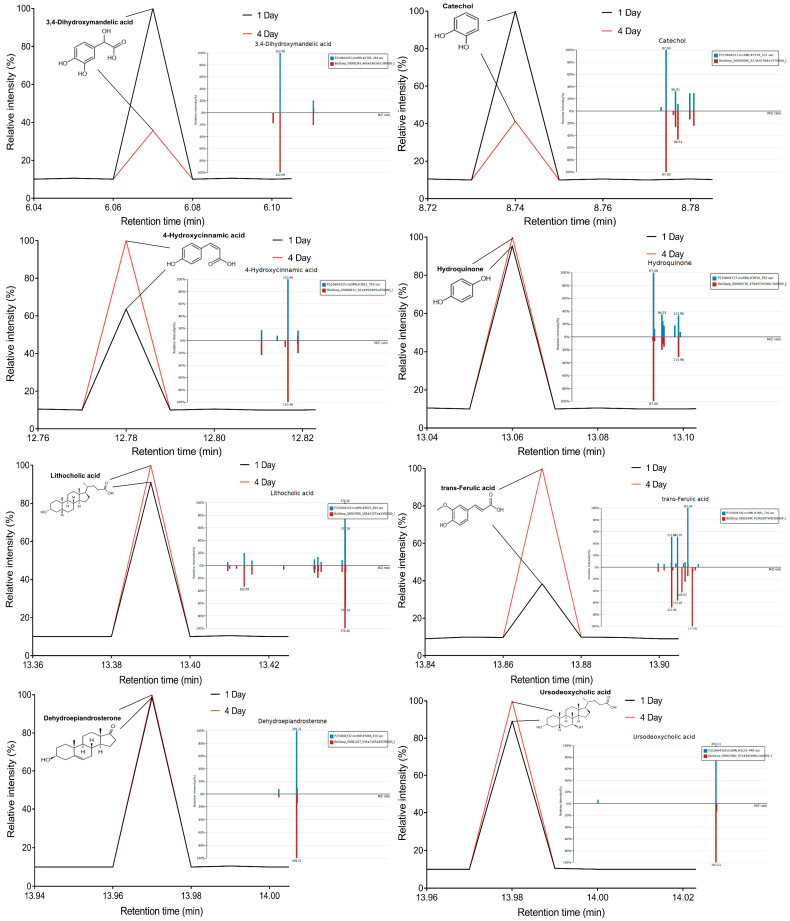
The chromatogram and mass spectrum of transformation products of bacterial E2 degradation.

**Figure 8 microorganisms-10-01745-f008:**
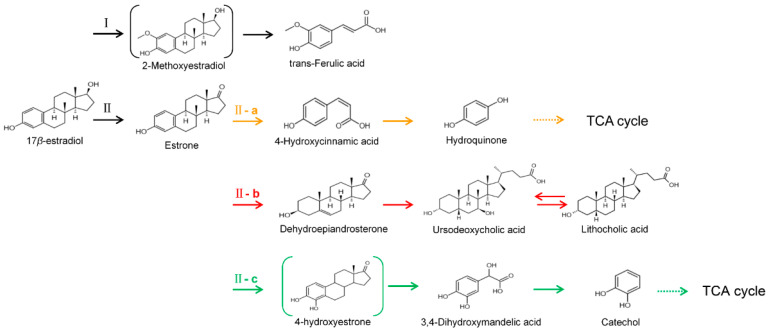
Proposed E2 transformation routes by bacterial E2 degradation. Products shown in the bracket were not identified in this study.

**Table 1 microorganisms-10-01745-t001:** Identification of and information about transformation products by bacterial degradation of E2 [15,16,17,18,19,20,21,22,23,24,25].

Chemical Names	Chemical Structure	Rt	*m*/*z*	Exacted Mass	Type	Elemental Formula	Reported by Others
3,4-Dihydroxymandelic acid	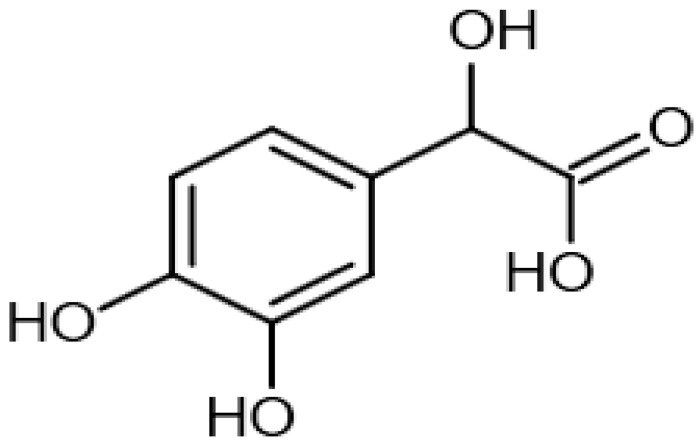	6.07	184.1692364	184.037	[M]^+^	C_8_H_8_O_5_	[21]
Catechol	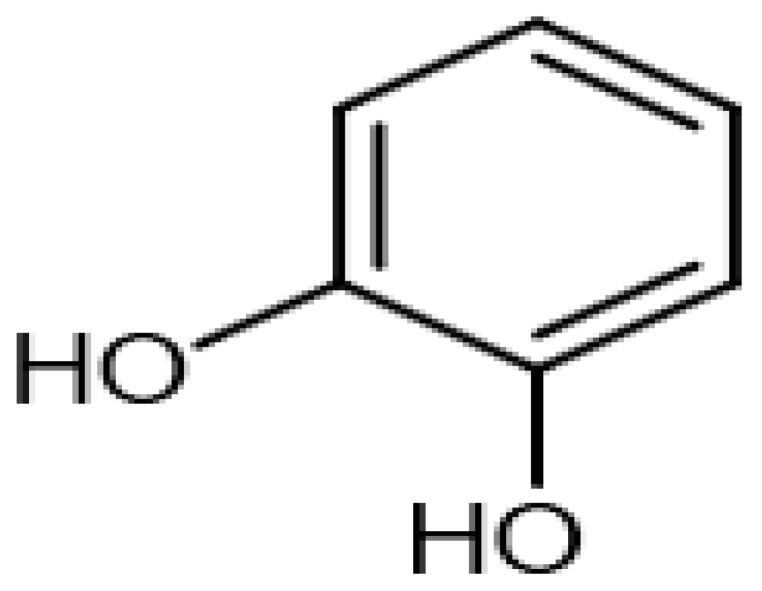	8.74	110.0210549	110.0368	[M]^+^	C_6_H_6_O_2_	[18]
4-Hydroxycinnamic acid	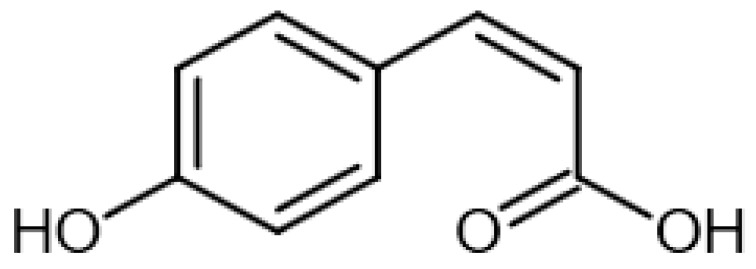	12.78	146.9804529	164.0473	[M+H-H_2_O]^+^	C_9_H_8_O_3_	[17]
Hydroquinone	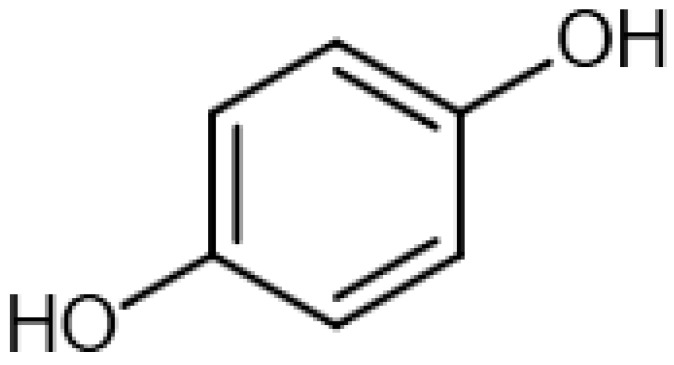	13.06	110.0208651	110.0368	[M]^+^	C_6_H_6_O_2_	[16]
Lithocholic acid	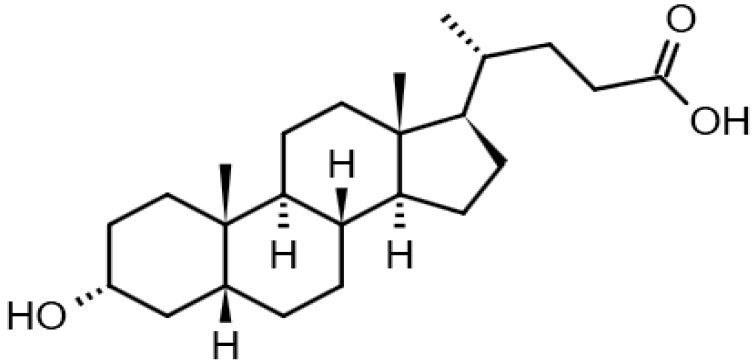	13.39	377.26 22	376.2977	[M+H]^+^	C_24_H_40_O_3_	[24]
trans-Ferulic acid	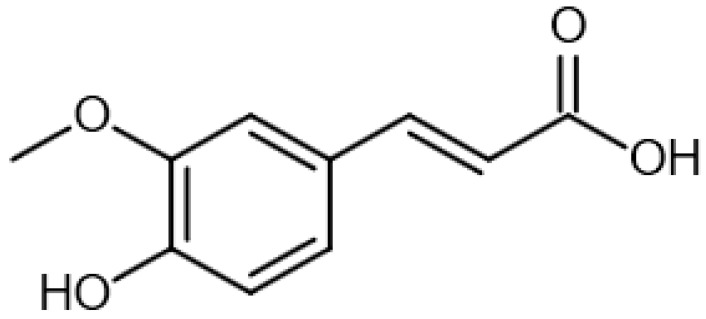	13.87	193.1232725	194.0579	[M-H]^−^	C_10_H_10_O_4_	[19]
Dehydroepiandrosterone	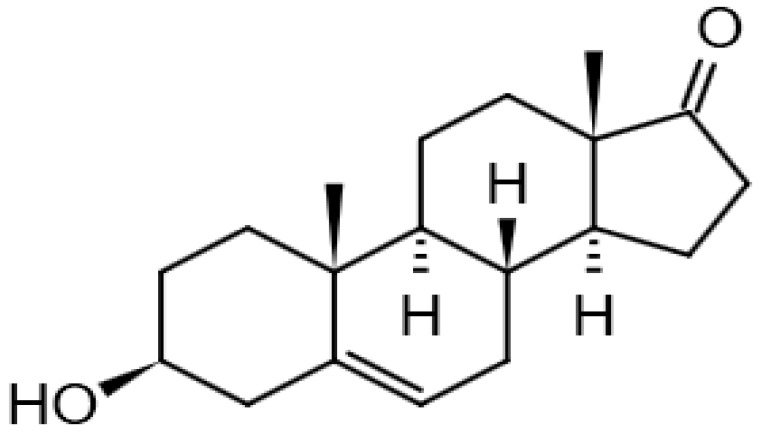	13.97	269.2142118	288.2089	[M-H_2_O-H]^−^	C_19_H_28_O_2_	[15,25]
Ursodeoxycholic acid	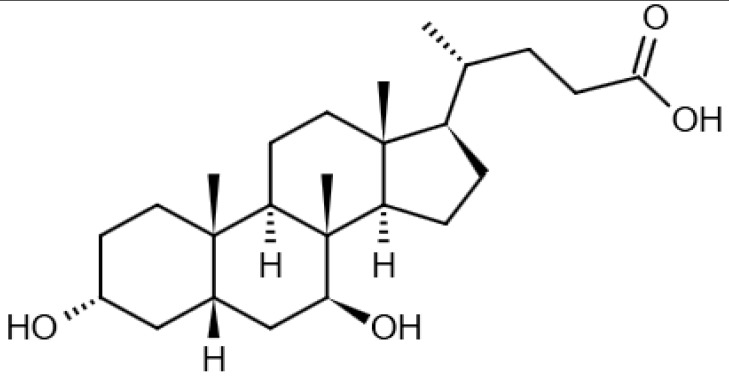	13.98	392.2858849	392.2927	[M]^+^	C_24_H_40_O_4_	[15,24]

Rt—Retention time. *m*/*z*—mass-to-charge ratio.

## Data Availability

The datasets generated during and analysed during the current study are available from the corresponding author on reasonable request.

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
