# Peer review of "Lysinibacillus sp. GG242 from Cattle Slurries Degrades 17β-Estradiol and Possible 2 Transformation Routes"

_microorganisms, 2022, doi:10.3390/microorganisms10091745_

Round 1

Reviewer 1 Report

Manuscript title:

Lysinibacillus sp. GG242 degrades 17β-estradiol and possible 2 transformation routes

I have carefully read the manuscript and found it interesting. However it cannot be published in current form as it requires a major revision.

11.  First of all OD600 can be effectively used for estimating bacterial growth only if soluble compounds are used as a sole carbon and energy source. In case of  compounds almost insoluble in water (as 17β-estradiol) there is no linear relationship at any range. The reason is that when growing on water insoluble substrates microorganisms form a biofilm on their surface.

22.   This why the results shown on Fig 4e are not reliable. The observed decreasing turbidity may as well be an indication, that more cells are attached to the surface of insoluble compound.

33.   As far as IC50  (or LD50) is concerned, it is also problematic to obtain a real values for a compound of that low solubility.  My advice is to omit these results from the manuscript.

44.   I would also like to know, what were the initial microbial numbers when the biodegradation experiment had started. OD600 values are quite useless for readers since the same values refer to different microbial numbers in case of different microorganisms.

55.   It is not clear how the Authors extracted the transformation products of  17β-estradiol from samples for LC-ESI-MS analysis.

66.   It is also not clear how extra- and intracellular enzymes were obtained. The cited reference is not freely available for most of the readers.

77.   What statistical test was used after ANOVA had shown statistically significant differences?

88.   It is quite tricky to deal with “%” instead of real values. For example 35% of 200 mg/l is more than 95% of 50 mg/l. So in terms of real values Lysinibacillus sp. GG242 was still a good degrader at higher concentration of 17β-estradiol. Please kindly consider presenting real values instead of %.

Reviewer 2 Report

I propose the editor of Microorganisms to accept your manuscript for publication, as the gained knowledge is of importance giving the long time and rising contamination of environments with estrogen.

However, I propose some changes and additions, to improve the paper

Title 2

 Lysinibacillus sp. GG242 from cattle slurries degrades 17β-estradiol and possible 2 transformation routes

Line 38 :  „estrogen in ng·L-1“ to be deleted or completed with concentrations.

Line 41; given example for inefficiency by applied methods or delete the word

Line 46 “which achieved a 99% removal rate of 1 mg·L-1 E2”

to be changed “

 which achieved in vitro a 99% removal rate of 1 mg·L-1 E2”

Lines 47 …; add also “in vitro”

Line 59: “was isolated” give the source. See also line 166, where the source must be given with more precision, as wastewater is unclear. Given cattle slurry if slurry is the origin as well as information if the slurry is fresh or fermented, or give more precision for waste water. Also for line 59

Line 166: give the reason why cattle “water” was chosen. High contamination with E2 and a problem with hygiene? High probability to isolate strains with capacity of degradation of E2?

Title Fig 2 - Add information about the culture medium as essential for morphology

Title Table 1 - Identification information of transformation products - Identification of and information about transformation products by bacterial degradation of E2

Title Fig 7 – add “of bacterial E2 degradation”

Title Fig 8 – add “by bacterial E2 degradation”

Line 267, Discussion.

The first sentence is not allowed in that form, as nothing is proven about activities of Lysinibacillus in environmental ecosystems. You investigated in culture systems, in vitro. The sentence should be adapted to the given experimental approach.

In line 303 you extrapolate “when faced with a real or harsh environment” To maintain the sentence, you should name here the conditions, the system used (in vitro and in culture or in situ).

In line 347 name also the original ecosystem of the strain.

Line 350: change “environmental adaptability“ into “cultural adaptability”

Line 354: “new ideas” are no perceptible in the manuscript. Biodegradation by microorganisms is not a new approach in case of degradation of organic pollutants at sites in situ, so former industrial areas, and is used in several sites worldwide.

The editor will take care that these comments are respected in the final version.

Round 2

Reviewer 1 Report

Now the manuscript is much improved. There is only one little issue, namely the initial number of bacterial cells was presumably 4.3·108 CFU·mL-1, rather than 4.3·1018 CFU·mL-1 (line 113).